# Blind Sequence Denoising with Self-Supervised Set Learning

## Abstract

Denoising discrete-valued sequences typically relies on training a supervised model on ground-truth sources or fitting a statistical model of a noisy channel. Biological sequence analysis presents a unique challenge for both approaches, as obtaining ground-truth sequences is resource-intensive and the complexity of sequencing errors makes it difficult to specify an accurate noise model. Recent developments in DNA sequencing have opened an avenue for tackling this problem by producing long DNA reads consisting of multiple subreads, or noisy observations of the same sequence, that can be denoised together. Inspired by this context, we propose a novel method for denoising sets of sequences that does not require access to clean sources. Our method, Self-Supervised Set Learning (SSSL), gathers subreads together in an embedding space and estimates a single set embedding as the midpoint of the subreads in both the latent space and sequence space. This set embedding represents the "average" of the subreads and can be decoded into a prediction of the clean sequence. In experiments on simulated long-read DNA data, SSSL-denoised sequences contain 31% fewer errors compared to a traditional denoising algorithm based on a multi-sequence alignment (MSA) of the subreads. When very few subreads are available or high error rates lead to poor alignment, SSSL reduces errors by an even greater margin. On an experimental dataset of antibody sequences, SSSL improves over the MSA-based algorithm on two proposed self-supervised metrics, with a significant difference on difficult reads with fewer than ten subreads that comprise over 75% of the test set. SSSL promises to better realize the potential of high-throughput DNA sequencing data for downstream scientific applications.

## 1 Introduction

Denoising discrete-valued sequences is a task shared by a variety of applications such as spelling correction (Angluin & Csűrös, 1997; Damerau & Mays, 1989; Mays et al., 1991), hidden Markov Model state estimation (Ephraim & Merhav, 2002), and biological sequence analysis (Tabus et al., 2002; 2003; Lee et al., 2017). In these settings, denoising typically involves specifying a statistical model of the noise process based on prior domain knowledge or training a supervised model on ground-truth source labels. Denoising biological sequences, however, presents unique challenges to both approaches. First, although supervised models (Figure 1b) have found some success (Baid et al., 2022), ground-truth source sequences are usually resource-intensive to obtain. Second, priors of self-similarity and smoothness that enable denoising in other domains such as images (Fan et al., 2019) often do not transfer to discrete, variable-length data. Third, assumptions on the independence of the noise process that underpin many methods are violated as the errors introduced during sequencing tend to be context-dependent (Abnizova et al., 2012; Ma et al., 2019).

Recent developments in DNA/RNA sequencing technology have highlighted the need for an efficient and accurate denoising method for biological sequences. Long-read sequencing platforms such as the Oxford Nanopore Technology (ONT) can process sequences of up to 2-5 million base pairs (bps), but come at the cost of much higher error rates (5-15%) compared to those of their predecessors (0.1-0.5%) (Amarasinghe et al., 2020). Many applications require these longer reads, such as the analysis of full single-chain variable fragments (scFv) of antibodies (Goodwin et al., 2016). In this sequencing paradigm, practitioners generate a long read consisting of multiple noisy repeated observations, or subreads, of the same source sequence. A denoising algorithm is then required to predict the clean sequence from the set of noisy subreads.

Traditional denoising algorithms involve analyzing $k$-mers (Manekar & Sathe, 2018; Yang et al., 2010; Greenfield et al., 2014; Nikolenko et al., 2013; Medvedev et al., 2011; Lim et al., 2014), fitting a statistical error model (Schulz et al., 2014; Meacham et al., 2011; Yin et al., 2013), or performing a multi-sequence alignment (MSA) of the subreads (Kao et al., 2011; Salmela & Schröder, 2011; Bragg et al., 2012; Gao et al., 2020). MSA-based algorithms (Figure 1a), the most commonly used, align the subreads and then collectively denoise them by identifying a consensus nucleotide base for each position (e.g. Kao et al., 2011). MSA-based denoising tends to be unreliable when very few subreads are available or high error rates lead to poor alignment. Even when MSA does not fail, it may not always be possible to break a tie among the subreads for a given position and identify an unambiguous "consensus" nucleotide.

In this paper, we propose self-supervised set learning (SSSL), a blind denoising method that is trained without ground-truth source sequences and can be applied even in settings where MSA produces poor results. SSSL, illustrated in Figure 1c, aims to learn an embedding space in which subreads cluster around their associated source sequence. Since we do not have access to the source sequence, we estimate its embedding as the midpoint between subread embeddings in both the latent space and sequence space. This "average" set embedding can then be decoded to generate the denoised sequence. Our SSSL model consists of three components, an encoder, decoder, and set aggregator. We formulate a self-supervised objective that trains the encoder and decoder to reconstruct masked subreads, trains the set aggregator to find their midpoint, and regularizes the latent space to ensure that aggregated embeddings can be decoded properly. Because SSSL is trained on the data and source sequences it will denoise, it can combine information across sets of subreads during training, allowing for more accurate denoising in few-subread settings.

We evaluate SSSL on two datasets, a simulated antibody sequence dataset for which we have ground-truth sequences, and a dataset of real scFv antibody ONT reads for which we do not. In settings without source sequences, we propose two metrics to evaluate and compare the success of our denoising method to other baselines. Our primary metric, leave-one-out (LOO) edit distance, upper bounds the edit distance from a denoised sequence to its associated source sequence. Our complementary metric, fractal entropy, measures the complexity of a denoised sequence relative to its subreads. On simulated antibody data, SSSL outperforms the MSA-based consensus baseline by an average source sequence edit distance of ∼12 bps, reducing errors by 31% and achieving strong results even on challenging reads with very few subreads. On real scFv antibody data, SSSL significantly improves LOO edit distance and fractal entropy relative to the MSA-based consensus baseline, particularly on small reads with fewer than 10 subreads which comprise over 75% of the test set. Denoising these reads enables their use for downstream analysis, helping to realize the full potential of long-read DNA platforms.

## 2 Background and Related Work

We first establish the notation and terminology used throughout the paper. Given a ground-truth source sequence $\boldsymbol{s} = (s_1, \cdots, s_{T_s})$ with length $T_s$, we define a noisy read $R$ as a set of $m$ subread sequences $\{\boldsymbol{r}_1, \cdots, \boldsymbol{r}_m\}$ with varying lengths $\{T_{\boldsymbol{r}_1}, \cdots, T_{\boldsymbol{r}_m}\}$. Each subread sequence $\boldsymbol{r}_i$ is a noisy observation of the source sequence $\boldsymbol{s}$ generated by a stochastic corruption process $q(\boldsymbol{r}|\boldsymbol{s})$. We refer to the percentage of tokens corrupted by $q$ as the *error rate*. In the case of DNA sequencing, this corruption process consists of base pair insertions, deletions, and substitutions. These errors are often context-dependent and carry long-range correlations (Ma et al., 2019). The error rates and profiles can also vary widely depending on the specific sequencing platform used (Abnizova et al., 2012).

### 2.1 DNA Sequence Denoising

The goal of DNA sequence denoising is to produce the underlying source sequence $\boldsymbol{s}$ given a noisy read $R$. Traditional denoising methods are based on $k$-mer analysis, statistical error modeling, or multi-sequence alignment (MSA) (Yang et al., 2012; Laehnemann et al., 2015). $k$-mer based methods (Manekar & Sathe, 2018; Yang et al., 2010; Greenfield et al., 2014; Nikolenko et al., 2013; Medvedev et al., 2011; Lim et al., 2014) build a library of aggregate $k$-mer coverage and consensus information across reads and use this to correct untrusted $k$-mers in sequences, but fail without large coverage of the same identifiable sections of the genome. Statistical error model-based methods (Schulz et al., 2014; Meacham et al., 2011; Yin et al., 2013) build an empirical model of the error generation process during sequencing using existing datasets, but require strict modeling assumptions and the fitting proecedure can be computationally expensive (Lee

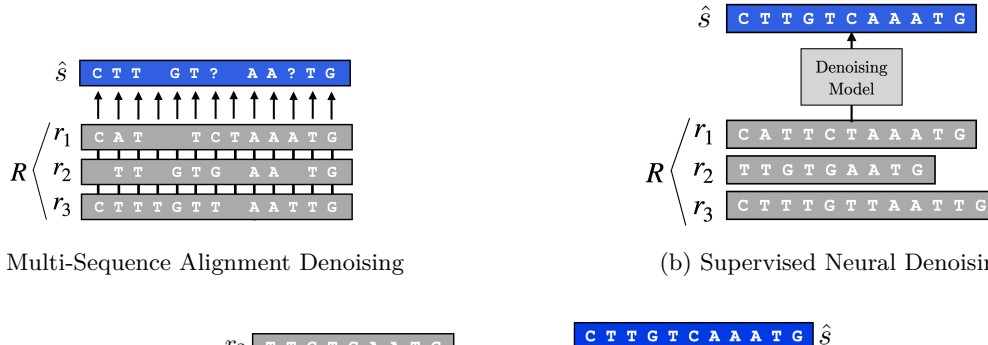

(a) Multi-Sequence Alignment Denoising      (b) Supervised Neural Denoising

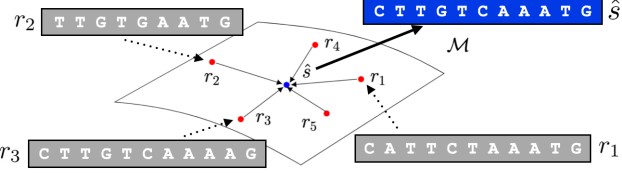

(c) Denoising with Self-Supervised Set Learning

Figure 1: Given a set of subreads $R = \{r_1, r_2, r_3 \cdots\}$, (a) commonly used denoising methods identify a per-position consensus from a multi-sequence alignment of the subreads. They are prone to failing when few subreads are available or error rates are high. (b) Supervised denoising methods train a neural network to directly predict the source sequence. During training, they rely on access to ground-truth sequences, which can be prohibitively expensive to obtain. (c) Our proposed self-supervised set learning framework denoises without access to source sequences by learning a latent space in which we can individually embed subreads, aggregate them, and then decode the aggregated embedding into a prediction of the clean sequence.

et al., 2017). MSA-based methods (Kao et al., 2011; Salmela & Schröder, 2011; Bragg et al., 2012; Gao et al., 2020) align the subreads within a read and then aggregate information by identifying a consensus nucleotide for each position, but perform poorly when few subreads are available or the subreads are noisy. Although combining these methods may alleviate some of these issues, current denoising methods are reliable only in limited settings.

Recent work has instead leveraged large neural networks to perform denoising by training a model to directly denoise reads in a fully supervised manner. Specifically, given a dataset $\mathcal{D} = \{(R^{(i)}, s^{(i)})\}_{i=1}^n$ consisting of reads $R^{(i)}$ generated from associated source sequences $s^{(i)}$, a neural denoising model learns to generate the source input $s^{(i)}$ given subread sequences in $R^{(i)}$ and any other associated features. Models like DeepConsensus (Baid et al., 2022) have shown that with a sufficiently large dataset of reads and ground-truth sequences, a model can be trained to outperform traditional denoising methods. However, ground-truth sequences are often prohibitively expensive to obtain, limiting the use cases for fully supervised neural methods.

## 2.2 Blind Denoising

In settings where we do not have access to ground-truth sequences, we need to perform *blind* denoising. Specifically, given a dataset that consists of only noisy reads, $\mathcal{D} = \{R^{(i)}\}_{i=1}^n$, we train a model that can generate the associated source sequences $s^{(i)}$ without ever observing them during training or validation. Existing work in blind denoising focuses on natural images, for which strong priors can be used to train models with self-supervised objectives. The smoothness prior (i.e., pixel intensity varies smoothly) motivates local averaging methods using convolution operators, such as a Gaussian filter, that blur out local detail. Another common assumption is self-similarity; natural images are composed of local patches with recurring motifs. Convolutional neural networks (CNNs) are often the architectures of choice for image denoising as they encode related inductive biases, namely scale and translational invariance (LeCun et al., 2010). In the case that the noise process is independently Gaussian, a denoising neural network can be trained on Stein's unbiased risk estimator (Ulyanov et al., 2018; Zhussip et al., 2019; Metzler et al., 2018; Raphan & Simoncelli, 2011). An extensive line of work ("Noise2X") significantly relaxes the assumption on noise so that it need

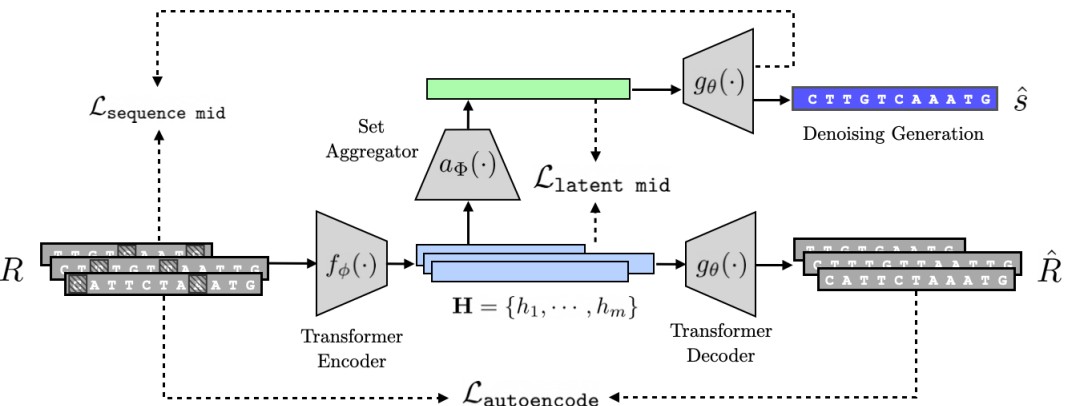

Figure 2: Our self-supervised set learning model architecture. All subread sequences in a given read first pass through a transformer encoder, which yields a set of embeddings. These embeddings are decoded by a transformer decoder and an autoencoding loss minimizes the reconstruction error. The set of embeddings are also combined by a set aggregator to produce a single set embedding, from which we can decode the denoised sequence using the transformer decoder. This set embedding is regularized to be the midpoint of the subreads in both the latent space and the sequence space.

only be statistically independent across pixels, allowing for a greater variety of noise processes to be modeled (Lehtinen et al., 2018; Batson & Royer, 2019; Laine et al., 2019; Krull et al., 2020).

Discrete-valued sequences with variable lengths do not lend themselves naturally to assumptions of smoothness, self-similarity, Gaussianity, or even statistical independence under the `Noise2X` framework—all assumptions explicitly defined in the measurement space. Moreover, we must model more complex noise processes for correcting DNA sequencing errors, which tend to be context-dependent and carry long-range correlations (Abnizova et al., 2012; Ma et al., 2019). We thus require more flexibility in our modeling than are afforded by common statistical error models that assume independent corruption at each position (Weissman et al., 2005; Lee et al., 2017).

## 3 Blind Denoising with Self-Supervised Set Learning

### 3.1 Motivation

Suppose we are given a dataset $\mathcal{D} = \{R^{(i)}\}_{i=1}^{n}$ of $n$ noisy reads, each containing $m^{(i)}$ variable length subread sequences $\{\boldsymbol{r}_j^{(i)}\}_{j=1}^{m^{(i)}}$ with correspond to noisy observations of a ground-truth source sequence $\boldsymbol{s}^{(i)}$. Assume that individual subreads and source sequences can be represented on a smooth, lower dimensional manifold $\mathcal{M}$ (Chapelle et al., 2006). Because we do not have direct access to $\boldsymbol{s}^{(i)}$, we cannot directly find its embedding in $\mathcal{M}$. However, since subreads from a given read are derived from a shared source sequence $\boldsymbol{s}^{(i)}$ and are more similar to $\boldsymbol{s}^{(i)}$ than to one another, we expect the representations of the subreads to cluster *around* the representation of their associated source sequence in $\mathcal{M}$, as shown in Figure 1c. In such an embedding space, we can estimate the embedding of $\boldsymbol{s}^{(i)}$ at the midpoint between subread embeddings in both the latent space and the sequence space. This embedding represents an "average" of the noisy subreads, and can then be decoded to generate $\hat{\boldsymbol{s}}^{(i)}$, our prediction of the true source sequence. We propose self-supervised set learning (SSSL) to learn the manifold space $\mathcal{M}$ accommodating all of these tasks: encoding individual sequences, finding a plausible midpoint by aggregating their embeddings, and decoding the aggregated embedding into a denoised prediction.

### 3.2 Model Framework

Our proposed SSSL model architecture is shown in Figure 2. It consists of three components:

- An encoder network $f_\phi$ parameterized by $\phi$ that takes as input a subread sequence $\boldsymbol{r}$ of $T_{\boldsymbol{r}}$ tokens and outputs a sequence of $d$ dimensional embeddings $\mathbf{h} \in \mathbb{R}^{d*T_{\boldsymbol{r}}}$.

- A set aggregator network $a_\Phi$ parameterized by $\Phi$ that takes as input a set of variable-length embeddings $\mathbf{H} = \{\mathbf{h}_1, \cdots, \mathbf{h}_m\}$ with $\mathbf{h}_i \in \mathbb{R}^{d*T_{r_i}}$, and outputs a set embedding $\hat{\mathbf{h}} \in \mathbb{R}^{d*T'}$ for some length $T'$.

- An autoregressive decoder network $g_\theta$ parameterized by $\theta$ that takes as input an embedding $\hat{\mathbf{h}} \in \mathbb{R}^{d*T'}$ and a sequence of previously predicted output tokens $(\hat{s}_1, \cdots, \hat{s}_{t-1})$ and outputs a distribution over the next token conditioned on the previous tokens $p(\hat{s}_t | \hat{s}_1, \cdots, \hat{s}_{t-1})$.

To denoise a given read $R = \{\boldsymbol{r}_1, \cdots, \boldsymbol{r}_m\}$, we pass each subread through the encoder to generate associated embeddings $\mathbf{H} = \{\mathbf{h}_1, \cdots, \mathbf{h}_m\}$. This set of variable-length embeddings is passed to the set aggregator to yield a set embedding $\hat{\mathbf{h}}$. The decoder then takes this set embedding $\hat{\mathbf{h}}$, representing the whole read, and generates a denoised sequence $\hat{\boldsymbol{s}}$. At no point during training or validation does our model observe the true source sequence $\boldsymbol{s}$.

In our experiments, we parameterize our encoder and decoder as a transformer encoder and decoder, respectively, with an additional projection head on top of the encoder network. Since the inputs to our set aggregator network are of varying lengths, we first transform the subread embeddings in the read to a common length by applying a monotonic location-based attention mechanism (Kim et al., 2021; Shu et al., 2019). The single transformed length $T'$ for a given read is calculated as the average of the subread lengths. The $\sigma$ scale is learned via a linear layer that takes as input a feature vector consisting of the $d$-dimensional average of the embeddings in $\mathbf{H}$, along with the sequence lengths $\{T_{\boldsymbol{r}_1} \cdots T_{\boldsymbol{r}_m}\}$. Once transformed to the same length $T'$, the subread embeddings are passed to a set transformer (Lee et al., 2019), a permutation invariant attention mechanism that combines the length transformed embeddings at each position across subreads to produce a single $\mathbb{R}^d$ embedding for each position and a sequence embedding $\hat{\mathbf{h}} \in \mathbb{R}^{d*T'}$. Specific model and training hyperparameters are provided in Appendix B.

### 3.3  Training Objective

We formulate a self-supervised training objective to construct an embedding space in which we can embed, aggregate, and decode sequences. The encoder $f_\phi$ and decoder $g_\theta$ are trained to map sequences to the embedding space and back to the sequence space. We optimize $f_\phi$ and $g_\theta$ with a simple autoencoding objective that attempts to minimize the log probability of reconstructing a given subread $\boldsymbol{r}_j$, i.e. encoding it with $f_\phi(\boldsymbol{r}_j)$ and decoding via $g_\theta$:

$$\mathcal{L}_{\texttt{autoencode}}(R) = -\sum_{j=1}^{m} \log g_\theta(\boldsymbol{r}_j | f_\phi(\boldsymbol{r}_j)). \tag{1}$$

We regularize the latent space learned by $f_\phi$ by applying a small Gaussian noise to the embeddings produced by $f_\phi$ and randomly masking the input subreads during training. In addition, we apply L2 decay on the embeddings to force them into a small ball around the origin. We call this regularization embedding decay:

$$\mathcal{R}_{\texttt{embed}}(R) = \sum_{j=1}^{m} \frac{1}{L_m} \sum_{k=1}^{L_m} ||f_\phi(\boldsymbol{r}_j)_k||_2^2. \tag{2}$$

The combination of these techniques acts similarly to those in LINDA (Kim et al., 2021), allowing the decoder decode properly not only from the embeddings of observed subreads but also from the aggregate set embedding at their center.

The set aggregator produces a single set embedding given a set of variable-length input embeddings. For convenience, we define $a_{\phi,\Phi}(R) = a_\Phi(\{f_\phi(\boldsymbol{r}_j)\}_{j=1}^m)$ as the set embedding produced by the combination of the encoder and set aggregator. In order to estimate the true source sequence embedding, we train the set aggregator to produce an embedding at the midpoint of the subreads in both the latent space and the sequence space. To find a midpoint in our latent space we require a distance metric $d$ between variable-length embeddings. We consider the sequence of embeddings as a set of samples drawn from an underlying

distribution defined for each sequence and propose the use of kernelized maximum mean discrepancy (MMD) (Gretton et al., 2012):

$$d_\kappa(\boldsymbol{x}, \boldsymbol{y}) = \left[ \frac{1}{L_x^2} \sum_{i,j=1}^{L_x} \kappa(\boldsymbol{x}_i, \boldsymbol{x}_j) - \frac{1}{L_x L_y} \sum_{i,j=1}^{L_x, L_y} \kappa(\boldsymbol{x}_i, \boldsymbol{y}_j) + \frac{1}{L_y^2} \sum_{i,j=1}^{L_y} \kappa(\boldsymbol{y}_i, \boldsymbol{y}_j) \right] \tag{3}$$

for a choice of kernel $\kappa$. In experiments we use the Gaussian kernel $\kappa(\boldsymbol{x}, \boldsymbol{y}) = e^{-||\boldsymbol{x}-\boldsymbol{y}||_2^2}$. Intuitively, this metric aims to reduce pairwise distances between individual token embeddings in each sequence while also preventing embedding collapse within each sequence embedding. To find a midpoint in sequence space we minimize the negative log likelihood of decoding each of the individual subreads given the set embedding. Combining these sequence and latent midpoint losses gives us the loss used to train our set aggregator:

$$\mathcal{L}_{\texttt{midpoint}}(R) = \sum_{j=1}^{m} \underbrace{- \log g_\theta(\boldsymbol{r}_j | a_{\phi, \Phi}(R))}_{\mathcal{L}_{\texttt{sequence mid}}} + \underbrace{d_\kappa(a_{\phi, \Phi}(R), f_\phi(\boldsymbol{r}_j))}_{\mathcal{L}_{\texttt{latent mid}}} \tag{4}$$

Putting all the components together, the objective used to jointly train the encoder, decoder, and set aggregator is

$$\arg\min_{\Phi, \phi, \theta} \frac{1}{\sum_{i=1}^n m^{(i)}} \sum_{i=1}^{n} \mathcal{L}_{\texttt{autoencode}}(R^{(i)}) + \eta \mathcal{L}_{\texttt{midpoint}}(R^{(i)}) + \lambda \mathcal{R}_{\texttt{embed}}(R^{(i)}), \tag{5}$$

where $\eta$ and $\lambda$ control the strength of the midpoint loss and regularization respectively. Rather than average losses over each read and then over each batch, we average over the total number of subreads present in a batch in order to ensure every subread in the batch is weighted equally. This weighting scheme has the effect of upweighing higher signal-to-noise reads with more subreads.

## 4 Evaluation Metrics

The ultimate goal for a given denoising method $f(\cdot)$ is to produce a sequence $f(R) = \hat{\boldsymbol{s}}$ from a given read $R = \{\boldsymbol{r}_1, \cdots, \boldsymbol{r}_m\}$ with the smallest edit distance $d_{\text{edit}}(\hat{\boldsymbol{s}}, \boldsymbol{s})$ to the associated source sequence $\boldsymbol{s}$. Since we do not observe $\boldsymbol{s}$, we require a self-supervised version of such a metric for evaluating and comparing denoising methods. We propose a primary metric, *leave-one-out (LOO) edit distance*, which upper bounds the edit distance $d_{\text{edit}}(\hat{\boldsymbol{s}}, \boldsymbol{s})$. We also propose a secondary metric, *fractal entropy*, which measures the entropy of a denoised sequence relative to its subreads. The combination of these metrics allows us to accurately evaluate and compare the performance of any denoising method.

### 4.1 Leave-One-Out Edit Distance

For our primary metric, we formulate a direct upper bound on the edit distance from our denoised sequence to the associated source sequence $d_{\text{edit}}(f(R), \boldsymbol{s})$. From the triangle inequality, we have

$$d_{\text{edit}}(f(R), \boldsymbol{s}) \le d_{\text{edit}}(f(R), \boldsymbol{r}_i) + d_{\text{edit}}(\boldsymbol{r}_i, \boldsymbol{s}). \tag{6}$$

Since $d_{\text{edit}}(\boldsymbol{r}_i, \boldsymbol{s})$ is a constant for a given subread, $\boldsymbol{r}_i$, $d_{\text{edit}}(f(R), \boldsymbol{r}_i)$ is an upper bound on the source edit distance $d_{\text{edit}}(f(R), \boldsymbol{s})$. However, we would like to measure the ability of the $f(\cdot)$ to denoise *unseen* sequences. We can instead denoise the set $R$ with $\boldsymbol{r}_i$ removed, $R_{-i}$. Our upper bound is then $d_{\text{edit}}(f(R_{-i}), \boldsymbol{r}_i)$ Averaging over all subreads, our final leave-one-out edit distance metric is:

$$LOO(R, f) = \frac{1}{m} \sum_{i=1}^{m} d_{\text{edit}}(f(R_{-i}), \boldsymbol{r}_i). \tag{7}$$

### 4.2 Fractal Entropy

Our secondary metric complements the LOO edit distance by quantifying how well a denoised sequence captures the patterns present in a set of subreads. Specifically, we measure the entropy of the denoised

sequence relative to its subreads. Intuitively, a denoised sequence should have a lower entropy when more of its $k$-mers are consistent with those present in the subreads, and higher when the $k$-mers in the denoised sequence are rare or not present in the subreads. Entropy has been applied in genomics for analyzing full genomes (Tenreiro Machado, 2012; Schmitt & Herzel, 1997) and detecting exon and introns (Li et al., 2019; Koslicki, 2011), and in molecular analysis for generating molecular descriptors (Delgado-Soler et al., 2009) and characterizing entanglement (Tubman & McMinis, 2012). However, directly calculating entropy for a small set of sequences with ambiguous tokens is difficult and often unreliable (Schmitt & Herzel, 1997).

We propose fractal entropy, a metric based on the Renyí quadratic entropy in a universal sequence mapping (USM) space estimated with a fractal block kernel that respects suffix boundaries. For a given sequence $\boldsymbol{r} = (r_1, r_2, \cdots, r_{T_{\boldsymbol{r}}})$, a USM (Almeida & Vinga, 2002) maps each subsequence $(r_1, \cdots r_i)$ to a point $s_i$ in the unit hypercube such that subsequences that share suffixes are encoded closely together. Details on the USM encoding process are provided in Appendix C. The USM space allows us to analyze the homology between two sequences independently of scale or fixed-memory context. To estimate the probability density of a sequence in this space, we use Parzen window estimation with a fractal block kernel (Vinga & Almeida, 2007; Almeida & Vinga, 2006) that computes a weighted $k$-mer suffix similarity of two sequences $s_i$ and $s_j$:

$$\kappa_{L,\beta}(s_i, s_j) = \frac{\sum_{k=0} L(|A|\beta)^k \mathbb{1}_{k,s_j}(s_i)}{\sum_{k=0}^{L} \beta^k} \tag{8}$$

where $\mathbb{1}_{k,s_j}(s_i)$ is an indicator function that is 1 when $s_i$ lies in the same section of the USM corresponding to sequences with the same length $k$ suffix as $s_j$. The kernel is defined by two parameters: $L$, which sets the maximum $k$-mer size resolution, and $\beta$, which controls the how strongly weighted longer $k$-mers are. For a given point $s_i$ with associated subsequence $(r_1, \cdots, r_i)$, the probability density can then be computed as:

$$\hat{f}_{L,\beta}(s_i) = \frac{1 + {}^1/_{T_{\boldsymbol{r}}} \sum_{k=1}^{L} |A|^k \beta^k c(r_i[:k])}{\sum_{k=0}^{L} \beta^k}. \tag{9}$$

where $r_i[:k]$ is the suffix of length $k$, $(r_{i-k+1}, \cdots, r_i)$, and $c(\cdot)$ is a count function that reduces the task of summing the indicator function values to counting occurrences of a particular $k$-mer in $\boldsymbol{r}$ (Almeida & Vinga, 2006). We can then calculate the Renyí quadratic entropy with this density estimate by integrating over the USM space, or equivalently by averaging the probability density over all possible $k$-mers of size $L$:

$$H_{L,\beta}^2(\boldsymbol{r}) = -\log \sum_{s \in A^L} \frac{1}{|A|^L} \hat{f}_{L,\beta}(s)^2. \tag{10}$$

To calculate the entropy of a given denoised sequence, we calculate $k$-mer frequencies for all $k \leq L$ in all subreads and the denoised sequence, and then compute the entropy as defined above. Since we want to measure the ability of a denoised sequence to aggregate information across multiple subreads, we use the entropy metric only on reads with at least 2 subreads, as the trivial lowest entropy denoised sequence for a single subread is a copy of the subread. When counting $k$-mers for MSA-based denoised sequences for which some $n \leq k$ ambiguous base pairs may be present, we add $^1/_{|A|^n}$ to the counts of each potential $k$-mer. For example, the $k$-mer `A?G` adds $^1/_4$ to the counts of $k$-mers [`AAG, ATG, ACG, AGG`]. We report results on using a kernel with parameters $L = 8$ and $\beta = 8$. Further discussion of specific kernel parameters and calculation are provided in Appendix C.

## 5 Experimental Setup

### 5.1 Data

**Sim Antibody:** We begin by investigating the ability of our model to denoise simulated antibody-like sequences for which the ground-truth source sequences are known. We generate a set of 10,000 source sequences **S** using a procedure that mimics V-J recombination, a process by which the V-gene and J-gene segments of the antibody light chain recombine to form diverse sequences. Details are provided in Appendix A. To generate the simulated reads dataset $\mathcal{D} = \{R^{(i)}\}_{i=1}^{n}$, we first select a sequence $s^{(i)}$ from **S** at random, then sample a number of subreads to generate, $m^{(i)}$, from a beta distribution with shape

parameters $\alpha = 1.3, \beta = 3$ and scale parameter 20. To mimic the worst error rates present in ONT sequencers (Laehnemann et al., 2015), we randomly select 15% of the base pairs in each of the $m^{(i)}$ subreads $\boldsymbol{r} \in R^{(i)}$ to corrupt. For each base pair selected, we insert a random base pair with 25% probability, delete the base pair with 50% probability, and substitute for a different base pair with 25% probability. We choose high deletion rates relative to the experimentally observed rates to test the ability of our model to accurately correct missing base pairs. Substitutions are made by selecting a different base pair uniformly at random. We generate a dataset of 100,000 reads and split these into a training, validation, and test set with a 90%/5%/5% split, respectively.

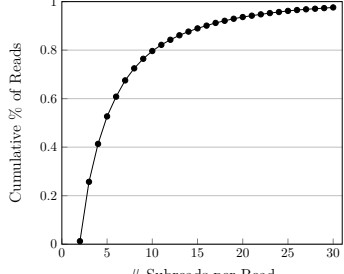

**scFv Antibody:** To investigate our model's ability to denoise real data, we use an experimental scFv antibody library sequenced with ONT. This dataset contains a total of 592,773 reads of antibody sequences, each consisting of a single light and heavy chain sequence. We focus on the light chains in this paper. Each read contains 2-101 subreads with an average of 8 subreads per read, and an average light chain subread length of 327 base pairs. The cumulative distribution of subreads per reads is shown in Figure 3. No reads of size 1 are included since we only include reads for which a valid consensus can be generated. As before, we split our data randomly into a training, validation, and test set by randomly sampling 90%/5%/5% of the data respectively.

Figure 3: Cumulative distribution of subreads per read in the scFv antibody test set. Over 75% of reads contain fewer than 10 subreads.

### 5.2 Baselines

We compare our method only against baseline methods that similarly only take the raw subread sequences as input. Our key baseline is an MSA-based consensus method, which generates a per-position consensus of the sub-reads from their alignment. We do not consider $k$-mer based methods as baselines in this paper, as they require access to genome alignments. We also do not consider methods based on statistical error modeling, which require extensive domain knowledge about the error-generating process. To perform the MSA, we use MAFFT v7.394 with default parameters (Katoh & Standley, 2013) and perform two rounds of alignment, then generate a consensus by selecting the relative majority base pair at each position. For positions where base pairs are tied we assign an "ambiguous" base pair. We refer to the generated sequence as the *consensus*.

## 6 Results

### 6.1 Sim Antibody

On our simulated data, we first examine how well SSSL denoises sequences compared to the MSA-based consensus baseline. Results are shown in Figure 4 and 5. On average, SSSL achieves a source sequence edit distance of 27.27 while the MSA-based consensus achieves a source sequence edit distance of 39.26, an improvement of nearly 12 bps. The distribution of edit distances, shown in Figure 4, demonstrates that our method not only improves the average edit distance to the source sequence, but also has lower variance. In Figure 5, our improvements over the MSA baseline are consistent regardless of the number of subreads per read, with the largest improvements occurring in the few-subread settings where alignment-based approaches struggle. SSSL is also able to denoise 1- and 2-subread reads, where MSA methods cannot be applied. Even for reads consisting of a single subread, SSSL achieves a lower edit distance than would be expected by simply copying the subread, indicating that the model has learned to aggregate information from other reads of the same source sequence.

Since we have ground-truth source sequences for each read, we also investigate the quality of our LOO edit and fractal entropy metrics. To evaluate the use of LOO edit distance as a proxy metric for source edit distance, we calculate the Pearson R (Freedman et al., 2007) correlation of these two values for both SSSL and consensus denoised sequences. We find a correlation of 0.83 for SSSL denoised sequences and a correlation of 0.78 for consensus denoised sequences, justifying our use of LOO edit for hyperparameter tuning and validation. To evaluate the use of both LOO edit distance and fractal entropy as a way to compare the quality of denoising methods on the same dataset, we calculate the Pearson R correlation between between

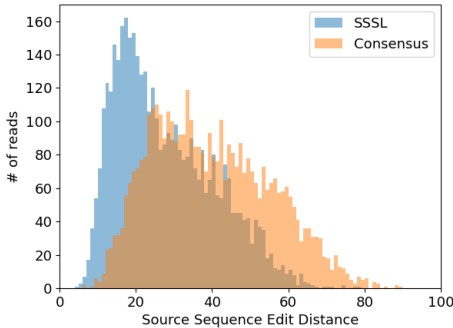

Figure 4: Edit distances from source sequences for SSSL and consensus predictions. SSSL outperforms the consensus by ∼12 bps on average with lower variance and a smaller range.

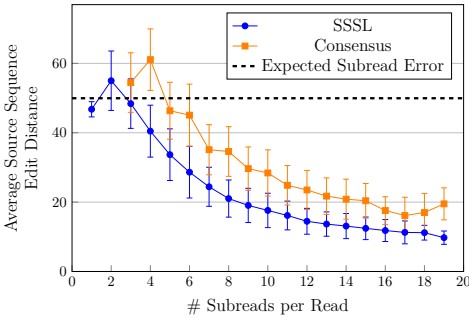

Figure 5: SSSL consistently achieves lower source sequence edit distance compared to the consensus across all subread sizes and can denoise even when there are only one or two subreads. Error bars represent standard deviations. The expected subread error is the expected edit distance between a subread and its source sequence.

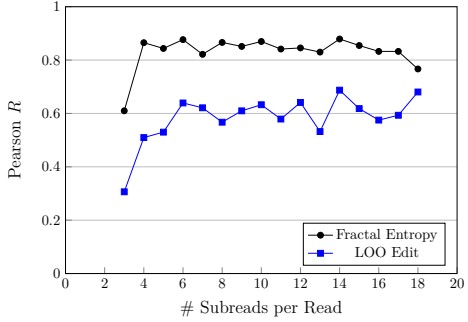

(a) Pearson correlation of differences in evaluation metrics with differences in source edit distance. Both metrics exhibit strong correlation across subread sizes.

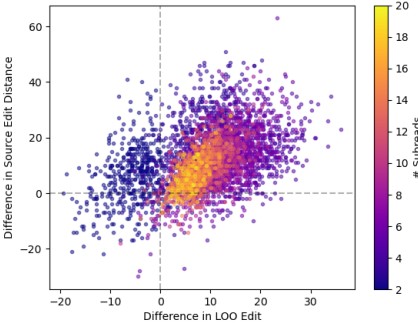

(b) As the number of subreads increases, differences in LOO edit exhibit a stronger linear relationship with differences in source edit distance.

Figure 6: Validation of LOO edit distance and fractal entropy as evaluation metrics.

differences in both metrics and differences in edit distance on a given read. Entropy is calculated with a fractal kernel of $L = 8$ and $\beta = 8$. Our analysis is presented in Figure 6.

We find that for reads of all sizes, differences in both LOO edit distance and fractal entropy correlate strongly with differences in source edit distance, with larger reads exhibiting strong correlation. When examining reads individually in Figure 6b, we find that reads with more subreads also align more closely with a line through the origin. These results motivate our use of both metrics as a way to compare the performance of different denoising methods in settings where we do not have ground-truth source sequences.

## 6.2 scFv Antibody

Results on antibody light chain data are shown in Figure 7. Since we do not have source sequences for the reads in this dataset, we analyze the differences in LOO edit and fractal entropy between consensus and SSSL denoised sequences. On average SSSL achieves a 21bp LOO edit distance while the consensus achieves a 26.2bp LOO edit distance, a 19.8% improvement of 5.2bp LOO edit. SSSL also achieves a 0.005 lower fractal entropy compared to the consensus. As the number of subreads decreases, the median difference for both

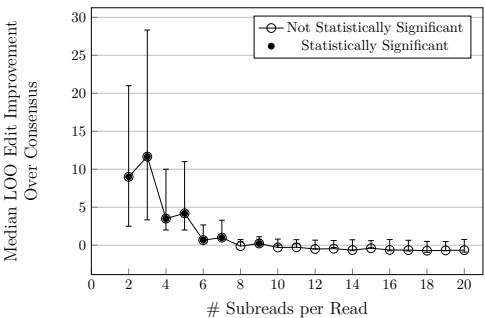
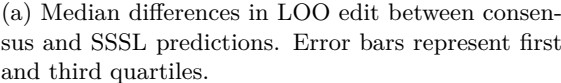

(a) Median differences in LOO edit between consensus and SSSL predictions. Error bars represent first and third quartiles.

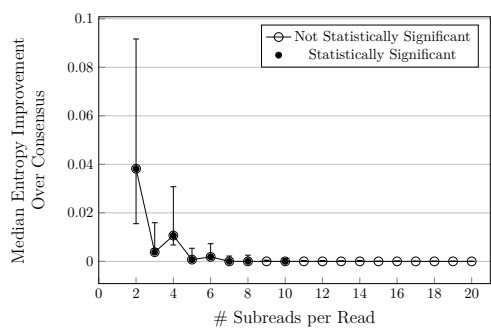

(b) Median differences in entropy between consensus and SSSL predictions. Error bars represent first and third quartiles.

Figure 7: Results on scFv antibody data. For both metrics, SSSL achieves significantly better performance on reads with fewer than 10 subreads, or over 75% of the test set. On average, SSSL outperforms the consensus by a LOO edit distance of 5.2 and entropy of 0.005.

metrics becomes significantly larger, with the largest difference of 11.7 LOO edit on reads with 3 subreads, and 0.038 entropy on reads with 2 subreads. This behavior is similar to that observed in simulation data, where SSSL improves most over the consensus on challenging reads where fewer subreads are present and performs more similarly as read size increases. For reads with less than 10 subreads, comprising over 75% of the test set, this difference is statistically significant. We measure statistical significance with a one-tailed paired t-test, a p-value of 0.005, and a null hypothesis that SSSL does not produce a better denoised sequence compared to the consensus.

The ONT platform is prone to introducing random insertions, deletions, and substitutions as well as homopolymer insertions, or erroneous repetitions of the same base pair. An ideal denoising algorithm is able to remove context-dependent errors as well as long homopolymer insertions, while maintaining true genetic variations present in the source sequence. We curate examples of SSSL and consensus denoised reads demonstrating each type of sequencing error in Figure 8. Here, "scaffold" refers to the sequence used at the library construction time to generate the read. While it is similar to the true source sequence, it may differ from it by some genetic variation.

In particularly noisy regions with homopolymers (Figure 8a), SSSL generates the correct number of base pairs while the MSA-based algorithm either generates too many or too few. In regions with many insertions and deletions (Figure 8b), SSSL properly removes inserted base pairs while the MSA-based algorithm cannot identify a consensus nucleotide and often outputs ambiguous base pairs. When real genetic variations are present in all the subreads (Figure 8c), both the consensus and SSSL produce the correct base pair, ignoring other reads it may have seen with different base pairs at those positions.

## 7 Analysis and Discussion

In this section we analyze how different data and model settings affect the downstream success of our denoising method as well as its ability to denoise unseen sequences. Since we can only control data parameters precisely on simulation data, all models are trained and evaluated on simulated antibody data generated from the same set of source sequences as described in Section 5.1.

### 7.1 Regularization

How important is each regularization component of our SSSL objective? SSSL has six key regularization components: sequence masking, embedding decay, embedding noise, sequence midpoint loss, latent midpoint loss, and $\eta$ loss weighting. To investigate the contribution of each component, we remove it and retrain the model for a fixed number of steps on the simulated antibody dataset. Results are shown in Figure 9. We find that all of our regularization components are important in improving the success of our model. The latent midpoint and sequence midpoint losses are the most important; removing them causes the model

```
SSSL      | tttgc  aaagtggggt
scaffold  | tttgc  aaagtggggt
consensus | tttgc aaaagtggggt
----------------------------
r001      | tttgc--aag---taagt
r002      | tttgc--aaaagtggggt
r003      | tttgcaaaaaagtggggt
r004      | tttgc---aaagtggggt
r005      | tttgc--aaaagtggggt
```

(a) Homopolymer insertions that throw off the alignment and present difficulties for MSA-based methods are denoised properly by SSSL.

```
SSSL      | tacttagcctggt accagcag...ggg acagac
scaffold  | tacttagcctggt accagcag...ggg acagac
consensus | tacttagcctgg ?ccagcag...ggg??cagac
-------------------------------------------
r001      | tacttagcctag  gctggcag...ggg-acagac
r002      | tacttagcctggtacccagcag...ggacggagac
r003      | tacttagcctgg  tacagcag...gggttcagac
```

(b) The MSA-based method outputs ambiguous predictions in sections with large numbers of insertions and deletions, whereas SSSL remains robust.

```
SSSL      | gaaattgtgttgacgcagtctccaggcaccctgtcttttgtc
scaffold  | gaaatagtgatgacgcagtctccagccaccctgtctgtgtc
consensus | gaaattgtgttgacgcagtctccaggcaccctgtctttgtc
----------------------------------------------
r001      |    agtgtgttgacgcagtctccaggcaccctgtcttttgtc
r002      | gaaattgtgttgacgcagtctccaggcaccc      tgtc
r003      | gaaattgtgttgacgcagtctccaggcaccctgtcttttgtc
r004      | gaaattgtgttgacgcagtct caggcaccctgtctttgtc
```

(c) Both methods are capable of preserving true genetic variations in the source sequence that differentiates it from the library scaffold.

Figure 8: Curated examples from the scFv dataset illustrating different types of errors.

to diverge and the loss to explode, so we do not display these results. $\eta$ loss weighting is the next most important, and increases the source edit distance on average by 9 bps, performing almost as poorly as the consensus. Removing masking increases the source edit distance by 6 bps, and removing embedding decay and embedding noise both perform similarly, increasing the source edit distance by 3 bps.

## 7.2 Subread Error Rates

How do different subread error rates affect the performance of SSSL denoising? We investigate error rates from 5% to 35% and generate a new dataset for each percentage using the same source sequences. We then train a model on each dataset until convergence. Results are shown in Figure 10. At the lowest error rate of 5%, SSSL performs similarly to the consensus. Within the range of error rates observed in long-read sequencing platforms ($\sim$10 - 25%), our denoising method significantly outperforms the consensus, with an improvement in edit distance of 3.5 bps at 10%, 12 bps at 15%, 12 bps at 20%, and 7 bps at 25%. Past 25% error, neither model is able to denoise better than the expected subread edit distance to the source sequence (i.e., denoising increases the existing noise present in the subreads).

## 8 Conclusion

We propose self-supervised set learning (SSSL), a method for denoising a set of noisy sequences without observing the associated ground-truth source sequence during training. SSSL learns an embedding space in which the noisy sequences cluster together and estimates source sequence representation as their midpoint. This set embedding, which represents an "average" of the noisy sequences, can then be decoded to generate a prediction of the clean sequence. We apply SSSL to the task of denoising long DNA sequence reads, for which current denoising methods cannot be applied when source sequences are unavailable or error rates are high. To evaluate our method, we propose two self-supervised metrics: leave-one-out (LOO) edit distance and fractal entropy. On a simulated dataset of antibody-like sequences, SSSL consistently outperforms our MSA-based consensus baseline. On an experimental dataset of antibody sequences, SSSL improves over the MSA-based consensus in terms of both self-supervised metrics. The difference is statistically significant on difficult reads with fewer than ten subreads, which comprise over 75% of the test set. By denoising these few-subread reads, SSSL enables their use in downstream analysis and scientific applications, more fully realizing the potential of long-read DNA sequencing platforms

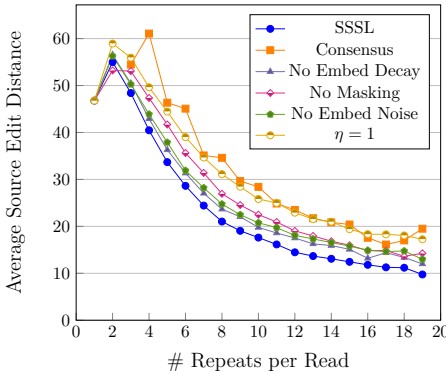

Figure 9: All our regularization techniques are important, and removing any individual component reduces the effectiveness of our method.

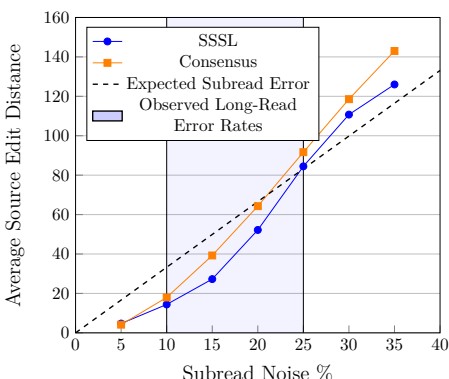

Figure 10: SSSL improves over the consensus at all except very low error rates (5%). The dashed line represents the expected edit distances between a given subread and its source sequence at a given error rate.

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

## A    Simulated Data Generation

In this section we describe our simulated data generation process. First, we randomly generate template V-gene and J-gene sequences by sampling a sequence length then randomly sampling a base pair from {A,T,C,G} uniformly and independently for each position. V-gene sequence lengths are sampled from a normal distribution with $\mu_v = 300$ and $\sigma_v = 6$. J-gene sequence lengths are sampled from a normal distribution with $\mu_j = 33$ and $\sigma_j = 3$. Given the resulting sets of V-gene and J-gene templates, $\mathbf{V}$ and $\mathbf{J}$, we then generate a set of source sequences $\mathbf{S}$ by concatenating each template sequence in $\mathbf{V}$ with each template sequence in $\mathbf{J}$. For our dataset we generate 100 $\mathbf{V}$ sequences and 100 $\mathbf{J}$ sequences for a total of 10,000 $\mathbf{S}$ sequences.

## B    Model and Training Hyperparameters

In this section we describe our model and training hyperparameters. We preprocess our data by tokenizing sequences using a codon vocabulary of all 1-, 2-, and 3-mers. We learn a token and position embedding with dimension 64. Our sequence encoder and decoder are 4-layer transformers (Vaswani et al., 2017) with 8 attention heads and hidden dimension of size 64. Our set transformer also uses a hidden dimension of size 64 with 8 attention heads. On top of the base encoder we apply an additional 3-layer projection head all with dimension 64 and BatchNorm (Ioffe & Szegedy, 2015) layers between each linear layer. Decoding is performed via beam search with beam size 32. All models are trained with the Adam optimizer (Kingma & Ba, 2014) with a learning rate of 0.001 and a batch size of 8 reads, although the total number of subreads present varies from batch to batch. We apply loss weighting values $\eta = 10$ and $\lambda = 0.0001$, and apply independent Gaussian noise to embeddings with a standard deviation of 0.01. Models and hyperparameters are selected based on validation LOO edit distance (Section 4.1).

## C    Fractal Entropy

### C.1    USM Encoding

Given an alphabet $A$ (for DNA, $A = \{$A,T,C,G$\}$), we define the universal sequence mapping (USM) space (Almeida & Vinga, 2002) in the $d = \log_2 |A|$ dimensional unit hypercube where each corner corresponds to a character in our alphabet. For a given sequence $\boldsymbol{r} = (r_1, r_2, \cdots, r_{T_{\boldsymbol{r}}})$ with length $T_{\boldsymbol{r}}$, we compute a sequence of USM coordinates $\mathbf{S} = \{s_1, s_2, \cdots, s_{t_{\boldsymbol{r}}}\}$ generated by a chaos game that randomly selects $s_0 \sim \text{Unif}(0,1)^d$ then calculates $s_i = 1/2(s_{i-1} + b_i)$ where $b_i$ is the corner of the hypercube corresponding to the character at position $i$.

### C.2    $L$ and $\beta$ Ablations

In this section we analyze the behavior of fractal entropy as our kernel parameters, $L$ and $\beta$ change. Intuitively, $L$ controls the resolution of the kernel, and the smoothing parameter $\beta$ control the weighting among $k$-mers of different lengths. Values of $\beta < 1/|A|$ correspond to higher weighting of shorter $k$-mers, and $\beta > 1/|A|$ correspond to higher weighting of longer $k$-mers (Almeida & Vinga, 2006). As we desire consistency between longer $k$-mers, we consider only values of $\beta > 1/|A|$. As $L$ increases, the correlation of fractal entropy increases as well, although large values of $L$ perform similarly. Since increasing the resolution of the kernel increases computation costs, we select a value of $L = 8$ as a reasonably high resolution. For $\beta \leq 1$, the correlation is poor, and for $\beta > 1$, the correlation is almost identical. Since the correlation values with $\beta = 8$ are marginally higher, we select this value in our experiments.

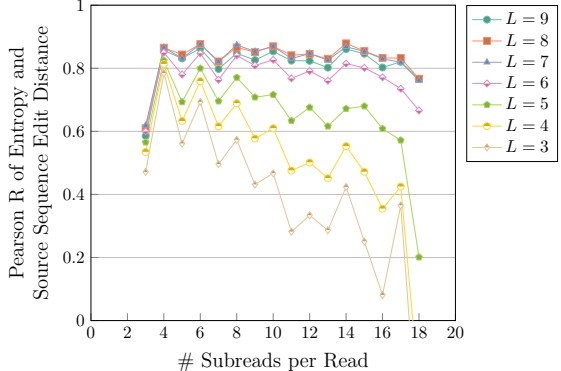
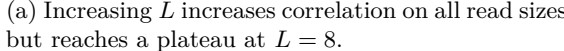
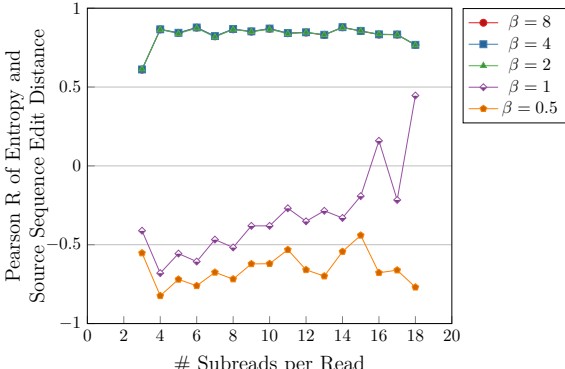

(a) Increasing $L$ increases correlation on all read sizes but reaches a plateau at $L = 8$.

(b) Values of $\beta \leq 1$ have poor correlation while values of $\beta > 1$ have almost identical correlation across all read sizes.

Figure 11: Ablations on fractal kernel parameters $L$ and $\beta$ which control kernel resolution and smoothness respectively.

