# OpenReview forum: "Blind Sequence Denoising with Self-Supervised Set Learning"
_TMLR — Rejected by TMLR_

### Review · Reviewer_fTvo · 2022-11-22

**Summary Of Contributions:**

This paper proposes a new set-based self-supervised embedding method for denoising reads (mainly of biological sequence data, such as DNA).

The key idea is to embed individual sub-reads in a shared embedding space in a self-supervised manner, aggregate them together, and then denoise by decoding from the aggregated embedding.

The method appears to improve performance compared to traditional MSA-based approaches on two biologically-relevant reconstruction datasets---both simulated and real-world.

**Audience:**

Yes

**Broader Impact Concerns:**

No concerns.

**Claims And Evidence:**

Yes

**Requested Changes:**

Following along the points I made in the previous section, I would request the authors to change their title to include the word 'biological'; e.g. _"Blind **biological** sequence denoising..."_.

Secondly, I would recommend adding baseline approaches that are parametric, i.e. not MSA-based. Perhaps the NeuroSEED approach I mention in the following paragraph could be appropriate, as it also learns a sequence-level encoder (although with a distance-based loss function)?

Thirdly, I believe a more thorough set of ablations is necessary to guide future use of the method. Specifically, the authors' method key component is the invariant set aggregation model. The authors currently mention they use Set Transformers and little more. Could they compare against a baseline that naïvely aggregates (by taking just a literal average of individual sequence embeddings, for example)? Or various choices of aggregation function (attention, sum, max, ...)? The authors may wish to consult either the set representation paper "On the Limitations of Representing Functions on Sets" (Wagstaff, Fuchs, Engelcke, et al., ICML'19) or Principal Neighbourhood Aggregation (Corso, Cavalleri et al., NeurIPS'20) when deciding on various aggregation schemes to use.

Lastly, w.r.t. positioning with related work: there have been many approaches designed to neurally handle biological sequences, especially under optimising the edit distance. One such paper is NeuroSEED (Corso et al., NeurIPS'21). The authors may find it useful to compare and contrast against this method (and any other relevant related methods) in their related work discussion.

I hope the above suggestions will be helpful to the authors when they prepare a revision.

**Strengths And Weaknesses:**

I have some bioinformatics experience and know that the problem studied by the authors is highly important and relevant. The approach proposed also seems timely and novel. However, I have no direct substantial experience relevant to the specific methodology being proposed, so my confidence in the method's novelty and the positioning against related work is not generally high.

In my opinion, the specific architecture consisting of embedding subreads, followed by decoding from their aggregate, is a solid idea likely to offer the kind of modelling generality necessary for this task.

That being said, the paper does have a few shortcomings which should be addressed (see 'Requested changes' for more details).

I feel like the title of the paper is too broad. While it is true that the methods proposed here could in principle be applicable to all kinds of discrete sequences, the main motivation and utility have been demonstrated on biological sequences. While this motivation is self-sufficient, I also think that biological sequences have very "special" input symbol distributions (e.g. C-G islands in DNA), and therefore any conclusions found on them would _not_ be naturally expected to carry over to generic sequences.

Secondly, I find the presented comparisons to be somewhat lacking. The authors only compare their method against one MSA baseline, and ablate their two regularisation hyperparameter choices. I feel that there is potentially insufficient insight for a practitioner wanting to use this method---especially given that this model has many moving parts (beyond the regulariser). Therefore I would recommend adding more ablations in the very least, and ideally including more relevant baselines (if any exist).

Lastly, I believe the work could benefit with slight enriching of its related work, to incorporate other (biological) sequence-based neural approaches.

---

> ### Author Response · Authors · 2022-12-12
> **Response to Reviewer fTvo**
>
> Thank you for the review. We respond to specific concerns below first, and will follow up in the coming week with results from additional experiments with other baselines and datasets as we complete them.
>
> **”I would request the authors to change their title to include the word 'biological'; e.g. "Blind biological sequence denoising...".”**
>
> We appreciate this suggestion and have updated our title accordingly.
>
> **"I would recommend adding baseline approaches that are parametric, i.e. not MSA-based. Perhaps the NeuroSEED approach”**
>
> We respond to these concerns in the general response (3) above.
>
> **“Could they compare against a baseline that naïvely aggregates (by taking just a literal average of individual sequence embeddings, for example)? Or various choices of aggregation function (attention, sum, max, ...)?”**
>
> We will add additional ablations on the aggregation method (set transformer, sum, max, average) as well as ablations on the choice of kernel for our kernelized MMD metric (gaussian, cosine). Please refer to our general response (4) above.
>
> **“I believe the work could benefit with slight enriching of its related work, to incorporate other (biological) sequence-based neural approaches.”**
>
> We plan to add additional related work on neural sequence representation learning methods. Please refer to our general response (2) above.

---

### Review · Reviewer_C7Np · 2022-11-25

**Summary Of Contributions:**

This paper proposes a method based on self-supervised learning to solve the problem of reconstructing objects based on several noisy observations of those objects, without requiring access to the original object. In this particular study, those objects are base pair sequences (DNA), since the problem of reconstructing DNA from several noisy samples is a common and unsolved problem in biology.

The method requires two components, an autoencoder and a set aggregator. The autoencoder is trained in a self-supervised fashion, and the set aggregator learns to encode sets in a useful way. In particular in this work, what the encoder and set aggregator produce are not fixed-dimensional embeddings, but rather variable-length sequences of embeddings. These sequences of encodings condition the autoregressive decoder.

As such there are three losses:
- A standard self-supervised reconstruction loss (of partially masked inputs)
- A "midpoint" loss, which pushes the aggregator to output an embedding that
    - minimizes maximum mean discrepancy between it and subread embeddings
    - minimizes the reconstruction error between it and subread embeddings
- A standard L2 decay on the embeddings

This paper also proposes to evaluation metrics that do not require access to the ground truth, in line with the motivating problem:
- A Leave-One-Out Edit-Distance that measures robustness to removing one subread from the set of subreads
- A "Fractal Entropy" measure, which quantifies the distributional discrepancy between the denoised sequence and the noisy subreads

Experiments are done on a synthetic dataset in which ground truth is known, and a real dataset where it isn't. The proposed method improves over a standard multi-sequence alignment algorithm.

**Audience:**

Yes

**Broader Impact Concerns:**

There is no impact section. I have no concerns on this work.

**Claims And Evidence:**

Yes

**Requested Changes:**

I think the paper would be much stronger with at least one of the questions I've listed in the Weaknesses above answered.

Other comments:

> attempts to minimize the log probability

Shouldn't this be "maximize"?

> randomly masking the input subreads

Figure 2 suggests that the autoencoding targets are the non-masked inputs, but equation (1) suggests otherwise. Perhaps writing $f_\phi(\tilde r)$ to make the noise explicit would be better.

> Since $d_{edit}(r_i, s)$ is a constant for a given subread, $r_i$, $d_{edit}(f(R), r_i)$ is an upper bound on the source edit distance $d_{edit}(f(R), s)$.

This doesn't seem correct. $a \leq b + c$, does not imply $a \leq b$, regardless of the value of $c>0$. $b$ is not an upper bound on $c$. Am I missing something? I don't understand the justification of the leave one out metric.

One thing that's missing is validating the the LOO and fractal entropy metrics correlate well not just on the predictions from SSSL, but from baselines as well.

**Strengths And Weaknesses:**

Strengths:
- The paper was well written and easy to understand
- The proposed method relies on well understood methods, leveraging self-supervised learning, and improves results on sensible benchmarks

Weaknesses:
- There is a lack of baselines. The authors justify this lack by discarding methods that require ground truth or domain knowledge, but these baselines would be especially interesting. What is [self-supervised] learning capturing here? What statistical regularities in the data are being picked up?
    - It's interesting that most of the gain measured by LOO-ED and entropy are coming from denoising sequences with very few subreads. Presumably what an ML-based model can do when it lacks sufficient information (too few subreads) is a "regression to the mean", but in a smart way, in a way which leverages patterns in the data, and so the blanks are filled in strategically. Could that be tested? Qualitatively or quantitatively, when comparing with methods that have, e.g. domain knowledge, are similar patterns recovered?
- Other than needing more baselines, the empirical evaluation is fairly well done, but at the same time leaves me unsatisfied:
    - It seems there's a gap between the effects observed in the synthetic task and the real task. Why? Could it have to do with structure in the noise? Is that something that could be tested with other [[semi-]synthetic] datasets?
    - Is this meant to be an application paper or a more fundamental methods paper? In the first case, I'd add more domain knowledge analysis in the paper, in the second case I'd add a totally unrelated (to biology) task that shows that this method is a capable "repeated noisy sample denoiser" method.

---

> ### Author Response · Authors · 2022-12-12
> **Response to Reviewer C7Np**
>
> Thank you for the review. We respond to specific concerns below first, and will follow up in the coming week on questions about additional experiments with other baselines and datasets as we complete them.
>
> **“There is a lack of baselines. The authors justify this lack by discarding methods that require ground truth or domain knowledge, but these baselines would be especially interesting.”**
>
> We respond to these concerns in the general response (3) above.
>
> **“What is [self-supervised] learning capturing here? What statistical regularities in the data are being picked up?”**
>
> Our self-supervised learning objective aims to cluster sequences from the same underlying read together, which implicitly combines aspects of both edit distance and closest string retrieval, since aggregated sequences must be close to their subreads in both sequence and latent space. However, since our objective does not rely on manually calculating either of these values it can be used to train neural embedding models on large scale datasets.
>
> **“Could [the model’s strategic leveraging of data] be tested? Qualitatively or quantitatively, when comparing with methods that have, e.g. domain knowledge, are similar patterns recovered?”**
>
> Although we cannot compare against models with domain or genome coverage knowledge due to our problem setting (see general response (2)), we plan to provide additional analyses on profiling the types of errors corrected by our method and other MSA-based methods.
>
> **“It seems there's a gap between the effects observed in the synthetic task and the real task. Why? Could it have to do with structure in the noise?”**
>
> We hypothesize that some of the performance gap is due to the way in which noise is introduced into the data during the sequencing process. Although we follow a similar process to [2] with uniform noise applied to each base with similar error rates for each error type, ultimately the forward corruption process is unknown. Exact knowledge of this process would allow for much higher quality denoising than is currently available.
>
> **I don't understand the justification of the leave one out metric.**
>
> We have removed erroneous deriviation of LOO edit as an upper bound. Additional information is provided in general response (5).
>
> **“Is this meant to be an application paper or a more fundamental methods paper?”**
>
> We believe this is a fundamental methods paper with a specific application to an important problem. To better position the scope of our paper relative to other work in the area we have updated the title to reflect our experiments in a specific biological context and our related work to better situate our work with respect to other neural approaches. More details are provided in general response (2).
>
> Specifically, our method proposes the first self-supervised objective for training sequence embedding models that does not rely on pairwise sequence-level supervision, allowing us to train on multiple orders of magnitude larger datasets compared to prior work. Our work provides a starting point for the development of more sophisticated set-based self-supervised objectives. In addition, we demonstrate that our method is able to leverage the learned latent space to share information across reads of the same sequence. We provide details on this additional analysis in our general response (6).
>
> [1] Gunjan Baid, Daniel E. Cook, Kishwar Shafin, Taedong Yun, Felipe Llinares-López, Quentin Berthet, Anastasiya Belyaeva, Armin Töpfer, Aaron M. Wenger, William J. Rowell, Howard Yang, Alexey Kolesnikov, Waleed Ammar, Jean-Philippe Vert, Ashish Vaswani, Cory Y. McLean, Maria Nattestad, Pi-Chuan Chang, and Andrew Carroll. Deepconsensus improves the accuracy of sequences with a gap-aware sequence
> transformer. Nature Biotechnology, Sep 2022. ISSN 1546-1696. doi: 10.1038/s41587-022-01435-7. URL https://doi.org/10.1038/s41587-022-01435-7.
>
> [2] Sahlin, K., Medvedev, P. Error correction enables use of Oxford Nanopore technology for reference-free transcriptome analysis. Nat Commun 12, 2 (2021). https://doi.org/10.1038/s41467-020-20340-8

---

### Review · Reviewer_gVqw · 2022-11-28

**Summary Of Contributions:**

This work presents a new algorithm for DNA sequence denoising based on self-supervised learning. Given a set of reads, a consensus denoised sequence is produced by encoding to a latent space, taking a mean in the latents space, then decoding a single denoised sequence. This is compared to a simple method of consensus on an MSA alignment. Experiments are performed with a focus on antibody sequencing both on a toy generated dataset and an antibody library sequenced with ONT. On the two defined metrics the proposed SSSL outperforms the consensus predictions for the i.i.d. train/val/test splits. Each component is shown to improve performance by an ablation study on the generated dataset. This work shows that a self-supervised learning method can help to denoise noisy sequencing reads without a reference genome.

**Audience:**

No

**Claims And Evidence:**

No

**Requested Changes:**

Requested Changes:

The contributions of this work are largely empirical, with the goal of showing SSSL’s usefulness over existing methods for denoising long reads without a reference genome. From the current results I do not fully believe that SSSL outperforms existing methods and would like to see more justification of this by either a deeper dive into on what data SSSL is particularly good at, or a wider empirical analysis over a larger diversity of real world datasets.

Critical:

- Results (in a similar fashion) on real sequencing data with ground truth reference genome.
- Comparison to other long read sequencing correction methods such as Nanocorr or ONCorrect or justification of why they are incomparable or not useful for this application.
- Stratified test splits i.e. evaluation on unseen sequences.

Would strengthen the work but are not critical:

- Further justification of the usefulness in downstream tasks for ML audiences.
- Deeper exploration in to the types of errors SSSL can catch which are not caught by MSA based methods.
- Exploration into the performance characteristics of SSSL in terms of sequence length, training time, sequence similarity to training set.
- Cleanup of entropy calculation and LOO metric justification.

**Strengths And Weaknesses:**

Strengths:

- Presents a (to my knowledge) novel architecture for aggregating sequences of differing lengths with multiple additional losses which improve performance.
- Method does not require  a reference genome or ground truth denoised sequences.
- While it is difficult to evaluate the models in this area due to the lack of ground truth sequences to test against, two measures are used and validated on synthetic data that do not require access to a ground truth reference genome.

Weaknesses:

- It is unclear to me exactly where the learning signal is coming from. What is the effect of a smaller training set size? Is it that SSSL is able to share information between reads on the same sequence?
- All experiments are done on a single? (unclear from the paper) 90/5/5 train/val/test split. Its unclear to me exactly how similar the train split is to the test split, but it seems that there could be substantial overlap in the sequences, especially in the synthetic “Sim Antibody” task based on the way that sequences were generated. From my understanding each gene (V,J) is seen over 100 times, in different combinations. Thus each gene in the test set has probably already been seen ~90 times. It seems unfair to compare this directly to a baseline that uses ~1/90 of the data. This leaves me wondering if it is possible to apply SSSL to new sequences? Or should SSSL be retrained on every dataset? It would be great to ensure that sequences in the test set are not the same as the ones in the training set through stratified splitting.
- This method is tested on real sequencing data but with no ground truth. It would be more convincing to show benefits on real data with a known ground truth, i.e. sequencing with some reference genome. I think this would improve validation over LOO and fractal entropy metrics.
- What kind of errors are caught by SSSL that are not caught by alignment-based methods? This is shown anecdotally in Figure 8, but it would be helpful to understand more quantitatively if there is a specific type of error that SSSL is more suited to correct.
- Could the authors add some justification to the simulated data generation process a bit? I’m not an expert in antibodies nor antibody sequencing. Why is this the “right” simulation procedure? And are there existing methods for simulating these sequences? Could we take a library of real sequences and add read errors to it instead? For instance, from my limited understanding, some genes have much more variation than others.

Questions:

How does this method compare to a specialized error correction method like Nanocorr or ONCorrect?

[https://genome.cshlp.org/content/25/11/1750.short](https://genome.cshlp.org/content/25/11/1750.short)

[https://www.nature.com/articles/s41467-020-20340-8](https://www.nature.com/articles/s41467-020-20340-8)

Could another performance metric be to leave out some reads from sequences with many reads and see how well the method performs on

Section 4.1: I do not follow the logic that “Since $d_{edit}(r_i, s)$ is a constant for a given subread, $r_i$, $d_{edit}(f(R), r_i)$ is an upperbound on the source edit distance $d_{edit}(f(R), s)$. This seems false. Example, let $f(R) = AA, s = GG, r_i = AG$ then $d_{edit}(f(R), s) = 2, d_{edit}(f(R), r_i) = 1$ and $d_{edit}(f(R), s) \nleq d_{edit}(f(R), r_i)$.

Could the authors clarify the statement “We focus on the light chains in this paper.” What does this mean practically? Are the heavy chain sequences discarded?

Why is the fractal entropy the right thing to do? I also found the section on it to be somewhat confusing. It would be helpful to understand first how a sequence is encoded as a distribution (which is currently in appendix c), and include a more complete explanation of the notation. I don’t understand how $\kappa_{L,\b\beta}$ is used. Where the $L$ in the numerator goes from eq. 8 to 9. I assume the sum in the numerator of eq. 8 is $k$ = 0 to L, this should be clarified. I assume $A^L$ is all possible sequences of length up to L?

How does sequence length affect the performance of SSSL? Transformer based architectures often have a difficult time with long sequences. Is this a concern for SSSL?

Out of curiosity, is there any interpretation of the R_{embed} regularization in probabilistic terms? This seems like a VAE type loss to me except with a fixed Gaussian noise. Pretty sure the KL between Gaussians with the same standard deviation is proportional to the L2 norm. This probabilistic interpretation might be interesting to explore further.

As someone not familiar with the application, it would be great to get a sense of how much an average edit distance improvement helps on downstream tasks. Is an edit distance decrease of 3.5bps at 10% critical? Nice to have? Irrelevant? What is a “usable” error rate? Any context here would be excellent motivation for a wider ML audience interested in learning more about this application.

Small things:

missing period at end of conclusion

figure 6(a) maybe should be “strong correlation across subread counts?” instead of sizes? sizes implies lengths to me.

Section 3.3 paragraph 1,  should be minimizing the **negative** log probability, or maximizing the log probability.

Section 4.2 $|A|$ is not defined in the main text.

---

> ### Author Response · Authors · 2022-12-12
> **Response to Reviewer gVqw**
>
> Thank you for the review. We respond to specific concerns below first, and will follow up in the coming week on questions about additional experiments with other baselines and datasets as we complete them.
>
> **“It is unclear to me exactly where the learning signal is coming from. What is the effect of a smaller training set size? Is it that SSSL is able to share information between reads on the same sequence?”**
>
> Yes, SSSL improves over MSA-based methods because it is able to share information between reads of the same sequence, or in the case of recombinant antibody sequences, between reads of sequences containing large sections of shared subsequences. In the worst case, consider a dataset of single reads per reference sequence where no large subsequences are shared. In this case there is no additional information that a neural method would be able to learn over a simple MSA-based method and so should not provide any improvement in our evaluation metrics.
>
> **“This leaves me wondering if it is possible to apply SSSL to new sequences? Or should SSSL be retrained on every dataset? It would be great to ensure that sequences in the test set are not the same as the ones in the training set through stratified splitting.”**
>
> When using our model to denoise, we do not have access to reference sequences and cannot build a stratified test split. To apply our method in these scenarios, hyperparameters should be chosen on a held-out validation set, then the final model should be trained on the entire dataset that we want to denoise.
>
> **“It would be more convincing to show benefits on real data with a known ground truth, i.e. sequencing with some reference genome.”**
>
> To our knowledge the only datasets with reference sequences are at the genome level [1, 2, 3] and not in settings with much shorter sequence lengths and higher depths that we focus on. Adapting these datasets for our method would require significant preprocessing work by a domain scientist. In addition, existing datasets for learning neural representations are extremely small scale. For example, the datasets used in the MSA experiments for NeuroSEED [4] contain a single read of a single sequence with 7000 subreads.
>
> We hope that the open sourcing of our simulated data and generation code provides a starting point for further investigation into our shorter sequence problem setting.
>
> **“What kind of errors are caught by SSSL that are not caught by alignment-based methods?”**
>
> Please refer to the general comment (6) for plans on additional analysis profiling the types of errors corrected by SSSL. We will update the general discussion with additional results once analysis is complete.
>
> **“Why is this the “right” simulation procedure? And are there existing methods for simulating these sequences? Could we take a library of real sequences and add read errors to it instead?”**
>
> The generation of our source sequences is closely models the process of VJ recombination specific to the antibody domain that we investigate. Our corruption procedure is very similar to the simulation procedure used to generate reads for ONCorrect [2], with uniform corruption applied across the sequence, and a slightly different deletion/insertion/substitution rate (50%/25%/25% in our work, 45%/20%,35% in [2]). Other methods for simulating these sequences such as NanoSim [5] rely on building profiles of error models from existing genome level reads which we do not have.
>
> **“How does this method compare to a specialized error correction method like Nanocorr or ONCorrect?”**
>
> We have added additional context and related work regarding our problem setting in general repsonse (2). Existing error correction methods like Nanocorr and ONCorrect focus in the genome level setting since coverage and reference sequences allow for high quality assemblies. Our work focuses on the second shorter sequence level case, for which fewer algorithms have been developed and methods like Nanocorr and ONCorrect cannot be applied.
>
> **“Could another performance metric be to leave out some reads from sequences with many reads and see how well the method performs”**
>
> To evaluate the generalizability of our model, we provide additional analysis on the per-source sequence edit distances compared to how many reads of those sequences were seen in the training set. Please refer to the general comment (6) for more discussion on this topic.
>
> **”I do not follow the logic [of the LOO edit upper bound]”**
>
> We have removed the erroneous deriviation of LOO edit as an upper bound. Additional information is provided in general response (5).

---

> ### Author Response · Authors · 2022-12-12
> **Response to Reviewer gVqw (continued)**
>
> **““We focus on the light chains in this paper.” What does this mean practically? Are the heavy chain sequences discarded?”**
>
> Both the light and heavy chains are sequenced as part of our scFv analysis, but there is a larger number of heavy chain scaffold sequences and thus fewer shared reads from the same ground truth sequences. We also expect these ground truth sequences to contain a much higher rate of genetic variation from their reference scaffolds compared to light chain sequences. For this reason we focus in this work only on denoising light chain sequences. We have updated the manuscript with this additional information.
>
> **“Why is the fractal entropy the right thing to do?”**
>
> In general there is no accepted way to quantify the “goodness” of an alignment or denoised sequence and is an open problem in the field. Although average edit distance to subreads is a common metric, we argue that it does not capture certain desirable aspects of a denoising algorithm that we would like to measure. Specifically, our fractal entropy metric allows us to measure k-mer motif similarity between sequences at multiple scales and positions. A similar metric was shown to recover known regulatory components and motifs from large genomes [6] and supports its use for measuring regularity between sequences. In contrast, edit distance based metrics compare positions individually and equally.
>
> **“It would be helpful to understand first how a sequence is encoded as a distribution (which is currently in appendix c), and include a more complete explanation of the notation. I don’t understand how $\kappa_{L, \beta}$ is used. Where the $L$ in the numerator goes from eq. 8 to 9. I assume the sum in the numerator of eq. 8 is $\kappa = 0$ to $L$, this should be clarified. I assume $A^L$ is all possible sequences of length up to L?”**
>
> A sequence is encoded as a distribution in sequence space via a process modeled with a chaos game as described in Appendix C. More intuitively for a given sequence $(r_1, r_2, \cdots, r_n)$, for each subsequence $(r_1, r_2, \cdots, r_k)$, we produce a point $s_k$ in our embedding space. The chaos game is designed such that all coordinates $\{s_i | r_i = a_0\}$ that share the same final character $a_0$ lie in the same section of the embedding space. This section can then be subdivided in the same way the full space was, such that all coordinates $\{s_i | r_i = a_0, r_{i-1} = a_1\}$ also lie in the same subsection. This fractal embedding structure motivates the fractal kernel $\kappa_{L, \beta}$ that we use. The fractal kernel distributes the probability mass of a given point $s_k$ among these subdivided sections depending on $L$, which controls how “deep” we subdivide, and $\beta$, which controls the weighting of each subdivided block.
>
> The $L$ in the numerator in eq. 8 was meant to be the upper bound of the summand not a term in the numerator. We have fixed this error in notation. $A^L$ is all possible sequence of length up to $L$.
>
> **“How does sequence length affect the performance of SSSL?”**
>
> In general, aligning longer sequences is more difficult since the larger number of insertions and deletions make frameshifts more severe. Our initial experiments with a base pair vocabulary showed that increasing the length of sequences past 400 slightly degraded the performance of SSSL due to difficulties in modeling long sequences. However, since we use a codon vocabulary that encodes 3bp at a time, the effective length of the sequences being modeled is reduced, allowing our model to handle up to 1.2k bp at a time without any degradation, and up to 3kbp with a small loss in performance.
>
> **“Out of curiosity, is there any interpretation of the R_{embed} regularization in probabilistic terms?”**
>
> Without the addition of the $R_{embed}$ regularization term, the decoder learns only to assign point probability mass in the latent space to the specific sequences seen during training. This prevents it from decoding properly from interpolated points between them. The addition of the embedding regularization is a common method in generative models that learn embedding spaces that admit decoding or sampling (most similar to our method are LINDA [7] and NeuroSEED [4] which both use similar regularization).
>
> **“Is an edit distance decrease of 3.5bps at 10% critical? Nice to have? Irrelevant? What is a “usable” error rate?”**
> An edit distance difference of 3.5bps at 10% changes the identity of an entire amino acid, and an edit distance difference of 12bp at 15% changes 4 amino acids. Since the antibody sequence-function landscape is peaky, denoising even a single amino acid improvement will improve the quality of downstream analyses.

---

> ### Author Response · Authors · 2022-12-12
> **Citations**
>
> [1] Goodwin, Sara, et al. "Oxford Nanopore sequencing, hybrid error correction, and de novo assembly of a eukaryotic genome." Genome Research 25.11 (2015): 1750-1756. https://genome.cshlp.org/content/25/11/1750.short
>
> [2] Sahlin, Kristoffer, and Paul Medvedev. "Error correction enables use of Oxford Nanopore technology for reference-free transcriptome analysis." Nature Communications
>
> [3] Gunjan Baid, Daniel E. Cook, Kishwar Shafin, Taedong Yun, Felipe Llinares-López, Quentin Berthet, Anastasiya Belyaeva, Armin Töpfer, Aaron M. Wenger, William J. Rowell, Howard Yang, Alexey Kolesnikov, Waleed Ammar, Jean-Philippe Vert, Ashish Vaswani, Cory Y. McLean, Maria Nattestad, Pi-Chuan Chang, and Andrew Carroll. Deepconsensus improves the accuracy of sequences with a gap-aware sequence
> transformer. Nature Biotechnology, Sep 2022. ISSN 1546-1696. doi: 10.1038/s41587-022-01435-7. URL https://doi.org/10.1038/s41587-022-01435-7.
>
> [4] Corso, Gabriele, et al. "Neural distance embeddings for biological sequences." NeurIPS 34 (2021): 18539-18551. https://arxiv.org/abs/2109.09740
>
> [5] Chen Yang, Justin Chu, René L Warren, and Inanç Birol; NanoSim: nanopore sequence read simulator based on statistical characterization. GigaScience, Volume 6, Issue 4, April 2017, gix010, https://doi.org/10.1093/gigascience/gix010
>
> [6] Vinga, S., Almeida, J.S. Local Renyi entropic profiles of DNA sequences. BMC Bioinformatics 8, 393 (2007). https://doi.org/10.1186/1471-2105-8-393
>
> [7] Yekyung Kim, Seohyeong Jeong, and Kyunghyun Cho. LINDA: Unsupervised Learning to Interpolate in Natural Language Processing. arXiv e-prints, art. arXiv:2112.13969, December 2021.

---

### Author Response · Authors · 2022-12-12
**General Response**

We apologize for the delay in responses due to the unexpected breakdown of the lead author’s laptop as well as the timing of the response period with NeurIPS. Accordingly, **we have requested an additional week to complete additional experiments and change our manuscript**. We appreciate the reviewers and action editor for remaining patient with us.

We thank the reviewers for their comments and are glad that they found our work novel (gVqw, fTvo), the problem setting important and timely (gVqw, C7Np, fTvo), and our paper well written and easy to understand (c7Np). Below, we describe requested changes that we **plan to incorporate** in the coming week and will follow up as we complete them:

1. **Title:** We will update our paper title to “Blind Denoising of **Biological** Sequences...” to better situate our investigation into the application of our method in biological specific contexts.
2. **Problem Setting and Related Work:** We will add additional context and related work that better situates our problem setting relative to existing work. Specifically, we describe two problem settings where long-read sequencing platforms such as ONT have been applied. First, in sequencing and assembling full genomes or transcriptomes on the order of millions of base pairs with fewer subreads and more coverage and overlap information. Second, in sequencing large libraries of shorter sequences on the order of thousands of base pairs with more variability in number of subreads and no overlap or coverage information. Existing datasets and work in alignment and denoising for long-read sequencing platforms such as Nanocorr [1] and ONCorrect [2] have focused on the first problem setting for which coverage and genome information is commonly available. In our work we focus on the second more difficult problem setting without such information where many reads can contain few subreads.
3. **Baselines:** We plan to add additional results on other baseline methods, including MSA-based approaches (Clustal [5], MUSCLE [6]), and Steiner string edit-distance minimization based approaches. We update our MAFFT results and all new baselines to perform only one round of alignment to better compare against SSSL. Other parametric neural embedding approaches (NeuroSEED [3], SENSE [4]) cannot be used in our problem setting since they rely on self-supervision from pairwise sequence-space metrics that cannot be calculated at the scale of our large datasets. For example, the existing datasets used for evaluation in NeuroSEED are small scale datasets containing only one read with 7600 subreads whereas our scFv dataset contains 600,000 reads with a total of over 2 million subreads. Since our method utilizes set-wise metrics without explicit calculation of any sequence-space metrics, it can be used at these dataset scales.
4. **Ablations:** We plan to add additional ablations of our model on the choice of aggregation method (max, average, sum, set transformer), as well as the choice of MMD kernel (Gaussian, cosine similarity, dot product).
5. **Evaluation Metrics:** We plan to add an additional evaluation metric, subread edit distance, or the average edit distance from a denoised sequence to each of its subreads. Our initial analysis shows that this metric displays strong correlation (~0.9 Pearson R) with ground truth source sequence edit distance on the simulated dataset. In addition, we remove erroneous analyses of our LOO edit metric and provide additional justification for the use of each of our 3 metrics, subread edit distance, fractal entropy, and LOO edit distance.
6. **Analysis:** We plan to add additional analysis of our results
- characterize the types of errors on scFv data where predictions made by SSSL and MAFFT differ considerably
- investigate how test set error changes depending on how many reads of a particular sequence were seen during training
- measure the performance of a model trained on simulated data with 15\% noise level on test data with different noise levels

We respond to each reviewer’s specific questions below and have also revised the manuscript to address the reviewers’ questions and comments. We thank the reviewers again for their time and are looking forward to a fruitful discussion period.

---

> ### Author Response · Authors · 2022-12-12
> **Citations**
>
> [1] Goodwin, Sara, et al. "Oxford Nanopore sequencing, hybrid error correction, and de novo assembly of a eukaryotic genome." Genome Research 25.11 (2015): 1750-1756. https://genome.cshlp.org/content/25/11/1750.short
>
> [2] Sahlin, Kristoffer, and Paul Medvedev. "Error correction enables use of Oxford Nanopore technology for reference-free transcriptome analysis." Nature Communications 12.1 (2021): 1-13. https://www.nature.com/articles/s41467-020-20340-8
>
> [3] Corso, Gabriele, et al. "Neural distance embeddings for biological sequences." NeurIPS 34 (2021): 18539-18551. https://arxiv.org/abs/2109.09740
>
> [4] Zheng, Wei, et al. "SENSE: Siamese neural network for sequence embedding and alignment-free comparison." Bioinformatics 35.11 (2019): 1820-1828. https://academic.oup.com/bioinformatics/article/35/11/1820/5140215
>
> [5] Ramu Chenna, Hideaki Sugawara, Tadashi Koike, Rodrigo Lopez, Toby J Gibson, Desmond G Higgins, and Julie D Thompson. Multiple sequence alignment with the Clustal series of programs. Nucleic acids research, 2003. https://pubmed.ncbi.nlm.nih.gov/12824352/
>
> [6] Edgar, Robert C. "MUSCLE: multiple sequence alignment with high accuracy and high throughput." Nucleic acids research 32.5 (2004): 1792-1797. https://pubmed.ncbi.nlm.nih.gov/15034147/

---

### Decision · Action_Editors · 2023-01-25

**Recommendation:** Reject

**Comment:**

The reviewers found the paper interesting and novel, but felt that the empirical evaluation was too weak, as it involved a single baseline, and one synthetic and one real-world dataset. They also felt that a more extensive discussion of related work was necessary in order to position the proposed method properly. While the reviewers were mostly satisfied with the changes promised in the author response, the authors did not upload a revised version of the paper, despite have been granted the requested extension.

The authors are encouraged to revise the paper, taking into account the reviewer suggestions, and re-submit.

**Audience:**

The reviewers agreed that there will be an audience for the paper, once the changes promised by the authors have been incorporated.

**Claims And Evidence:**

The reviewers found an issue with only one of the claims in the paper: the claim used to motivate the Leave-One-Out edit distance metric, namely that it is an upper bound on the edit distance of interest, is false.